# Select the Key, Then Generate the Rest: Improving Multi-Modal Learning with Limited Data Budget

## Abstract

Multimodal learning serves as a promising approach for applications with diverse information sources. However, there are numerous challenges when scaling up multimodal learning data from all modalities due to availability or varied cost of data collection. We are the first to demonstrate that multimodal models with only a subset of modalities available for new data could reach and even surpass models continuously trained with full modalities. Our research problem is formulated as: *given a limited data collection budget, how to find the appropriate modalities to collect new data and generate for the rest to maximize model performance gain?* To answer this, we propose a new novel paradigm - **S**elect the **K**ey modality, then generate the rest to enable **l**earning with **l**imited data (`SK-ll`). `SK-ll` contains two essential components: (1) *Select the key*. We propose a modality importance indicator to find the optimal modalities by assessing their single modal marginal contribution and cross-modal interactions. (2) *Generate the rest*. We substitute with generated embeddings for the rest of modalities, . We conducted extensive experiments applying `SK-ll` across affection computing, healthcare with diverse multimodal learning backbones, obtaining average accuracy gains of {2.37%, 6.55%, 6.73%} on {`MOSI`, `MOSEI`, `ADNI`} respectively. Meanwhile, we present interesting empirical insights such as the data efficiency. Codes are provided in the supplement material.

## 1 Introduction

Multimodal learning (MML) integrates information from different modalities, such as text, visual, sensor, biomarkers to learn implicit representation, which has been applied to diverse areas including visual question answering (Ilievski & Feng, 2017; Ding et al., 2022; Lu et al., 2023), sentiment analysis (Soleymani et al., 2017; Chen et al., 2017; Gandhi et al., 2023), robotics (Sun et al., 2021a), and healthcare application (Yun et al., 2024). Despite its great potential in these applications, collecting data with full modalities in real-world scenarios can be challenging and costly. This is due to restrictions in real life, for example in biomedical settings, measurement devices would destroy paired samples (Xi et al., 2024). In addition, the cost of collecting different modalities varies significantly, with easily accessible image-text data being far more abundant than more complex modalities such as depth or thermal maps (Zhu et al., 2024; Girdhar et al., 2023), tactile data requires specialized sensors (Zou et al., 2017; Yang et al., 2017; Meribout et al., 2024). Previous studies have devised various approach on multimodal learning with missing modalities (Qiu et al., 2023; Wang et al., 2023a; Wu et al., 2024; Lee et al., 2023a; Guo et al., 2024). However, the missing case is created on full available data by randomly dropout, some even utilize the dropped modality data for reconstruction learning (Wang et al., 2023a), thus does not reflect reality. In sum, there is still no comprehensive solution to MML under limited data.

We conducted preliminary experiments over a few affection computing datasets (Zadeh et al., 2016; 2018) to show the importance of multimodal learning, while the contribution of different modality combinations differs substantially. We first investigate the efficiency of data utilization of multi-modal learning compared to unimodal learning. This is experimented with evaluating the performance of a multimodal transformer (Yu et al., 2019) across different training data ratios against the performance of the best unimodal transformer model trained with the complete dataset. As illustrated in Figure 1 (Upper) for both the `MOSI` and `MOSEI` affection computing datasets, the models trained with full

multimodality surpass the best unimodal baseline even when using as little as 10% of the total training data. However, not all modalities or their combinations contribute equally to the performance of multimodal learning. To further assess the potential of utilizing a subset of modalities, we trained multimodal models with various combinations of the available modalities. Their test accuracies were then compared to a model trained on the full set of modalities. As depicted in Figure 1 (Bottom), specific subsets of modalities (indicated by the green line) can achieve strong performance relative to other subsets, and in some cases, can approach the performance of the model trained with all modalities (indicated by the orange line). For example, in the MOSI dataset, the Audio+Text combination is a notably high-performing subset. For the MOSEI dataset, combinations such as Video+Text and Audio+Text also yield strong results. The observation that the most effective modality subsets can vary by dataset highlights that a strategic selection of modalities is crucial. This approach can lead to superior or more efficient performance, particularly when data acquisition or processing resources are constrained, potentially without needing a complete set of all available modalities.

Inspired by the findings from the preliminary experiment and data collection difficulty in reality, we introduce a new research problem: *given a certain limited data collection budget, how to determine which modalities should be collected and the rest should be generated for improved multimodal learning?* To address this question, we propose a novel multimodal paradigm, **S**elect the **K**ey, then generate the rest to enable **l**earning with **l**imited data (SK-ll). Specifically, SK-ll consists of two key components: the modality importance indicator and the missing modality generation.

Firstly, the modality importance indicator selects the key modality combination by evaluating both the marginal contribution of each modality and their cross-modality interactions. Given a multimodal dataset, we apply our proposed Step-wise Maximization of Modality Selection algorithm to identify the most informative modality combination. We then allocate the data collection budget to the key combination to ensure that the most informative modalities are prioritized. Secondly, we generate the missing modalities based on the combination to ensure the completion of multimodal data while mitigating modality interference. This approach

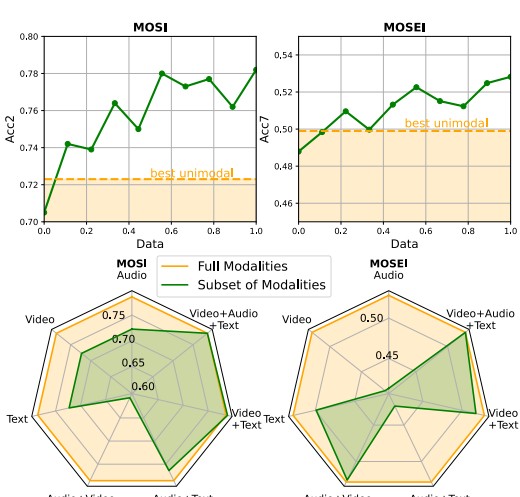

Figure 1: Top: Accuracy of multimodal model at different training data ratios (0.1 to 1.0) across affection computing datasets, compared to best unimodal learning (orange area) at full training data. Bottom: Test accuracy of multimodal models learned on different modality combination (green line) compared to the model trained on full modalities (orange line) across 2 affection computation datasets. Acc2 denotes binary accuracy, while Acc7 denotes 7-class accuracy.

enables more effective model training even with a limited data collection budget. To validate the effectiveness of SK-ll, we conduct experiments across MML backbones and applications. The contributions of this paper are summarized as follows:

• We have conducted pioneering investigation into multimodal learning under limited data collection budgets, addressing the practical constraint of varying modality acquisition costs. We reveal that MML is more data-efficient than unimodal approaches, while performance varies significantly across modality subsets, necessitating principled modality selection for achieving optimal MML performance under budget constraints.

• We introduce a novel framework that first Select the Key modality subsets for new data collection, then generate for the rest modalities, thus enable Learning with Limited new data. Then, We develop indicator for assessing the relative importance of each modality subset, which guides modality selection in SK-ll. Experiments demonstrate a strong correlation between the rankings of this indicator and the accuracy of actual MML task across different subsets of modality.

• Our framework offers flexibility by adapting to any available data at any new data budget constraint, enhancing performance across diverse MML tasks. We validate the effectiveness of SK-ll through extensive experiments across multiple multimodal backbones. Our approach demonstrates consistent

performance improvements across affective computing and healthcare. For example, SK-ll obtains average accuracy gains of {2.37%, 6.55%, 6.73%, 0.59%} on {MOSI, MOSEI, ADNI, MIMIC} compared to baseline approach.

## 2 RELATED WORK

**Multimodal Learning with Missing Modality.** Real-world multimodal systems often face the challenge of uncertain modality missingness due to various factors such as environmental interference, sensor malfunctions, and privacy concerns, which can significantly degrade model performance. Consequently, developing robust MML models that can effectively handle missing modalities has become a critical focus in the field (Ma et al., 2022; Wei et al., 2023; Lee et al., 2023b; Qiu et al., 2023; Zhang et al., 2023b; Wu et al., 2024). Recent studies have further explored modality robustness and unified frameworks to mitigate these effects. For instance, Hazarika et al. (2022) and Lin & Hu (2023) analyze the robustness of sentiment analysis models against modality drops, while UniMF (Huan et al., 2023) proposes a unified framework for unaligned and missing sequences. Techniques to address this range from simple imputation (Tran et al., 2017; Pham et al., 2019; Wang et al., 2023b) to sophisticated strategies like noise imitation (Yuan et al., 2023) and multimodal mixup (Lin & Hu, 2024) to enhance representation robustness. Other approaches utilize deep generative models to synthesize either the missing data itself or its latent representation (Hoffman et al., 2016; Zheng et al., 2021; Zhou et al., 2021; Zhi et al., 2024). Huang et al. (2025) push this boundary by exploring out-of-modal generalization without relying on instance-level modal correspondence. However, these studies predominantly focus on the retrospective problem of handling missing values or ensuring robustness within a fixed, pre-existing dataset. In contrast, we focus on the critical, practical constraint of a limited data collection budget. Our work addresses the prospective challenge of creating a systematic methodology to prioritize which modalities to collect initially, aiming to maximize performance gain from new data acquisition.

**Modality Imbalance and Selection in MML.** By learning complementary information from multiple sources, it is expected that MML can achieve better performance than using a single modality. However, recent works have shown that some modalities are more dominant than others (Du et al., 2023; Peng et al., 2022), and different modalities overfit and converge at different rates (Wang et al., 2020), leading to the modality imbalance problem and counterproductive MML performance (Ismail et al., 2020; Sun et al., 2021b). Fan et al. (2023) found that the dominant modality not only suppresses the learning rates of other modalities but also interferes with their update direction. Several methods have been proposed to address this, such as modulating learning pace (Zhang et al., 2023a), alternating unimodal learning (Zhang et al., 2024), or using sparse mixture-of-experts (Peng et al., 2023). Wei & Hu (2024) propose boosting multimodal learning via innocent unimodal assistance to achieve Pareto optimality. Yang et al. (2024) facilitate classification by dynamically learning and bridging the modality gap. While gradient conflicts have been studied to mitigate modality collapse (Javaloy et al., 2022), the selection of optimal modalities remains a challenge. Recently, He et al. (2024) proposed selecting modalities based on Shapley values to improve inference efficiency. However, their work focuses on reducing computational FLOPs during inference by pruning modalities, whereas our work focuses on the *data acquisition* stage. We aim to optimize the collection budget by selecting synergistic modalities and generating the rest, thereby addressing both the cost of acquisition and the issue of detrimental modality interference.

## 3 METHODOLOGY

### 3.1 TASK FORMULATION OF MULTIMODAL LEARNING UNDER LIMITED DATA BUDGET

In this section, we describe the resource-constrained multimodal learning problem in which we have a limited data collection budget $B$ in addition to available data. Formally, let $\mathcal{M} = \{M_1, \ldots, M_{|\mathcal{M}|}\}$ be a set of $m$ modalities, e.g., video ($V$), audio ($A$), and text ($T$). Let $\mathcal{D} = \{(x_1, y_1), \ldots, (x_{|\mathcal{D}|}, y_{|\mathcal{D}|})\}$ denote the available training data, where refers to the number of samples in the dataset. The input $x_i$ belongs to the input space $\mathcal{X}$, and the output $y_i$ belongs to the output space $\mathcal{Y}$. Specifically, $\mathcal{Y}$ can represent a finite set of discrete classes for classification tasks or the space of possible output sequences for generation tasks. We further assume that each input $x_i$ can be decomposed into components corresponding to each modality: $x_i = x_i^1, \ldots, x_i^{|\mathcal{M}|}$, where $x_i^j$ indicates the data for the modality $M_j$ associated with the $i$-th sample. Given a fixed data collection budget $B = M * N$.

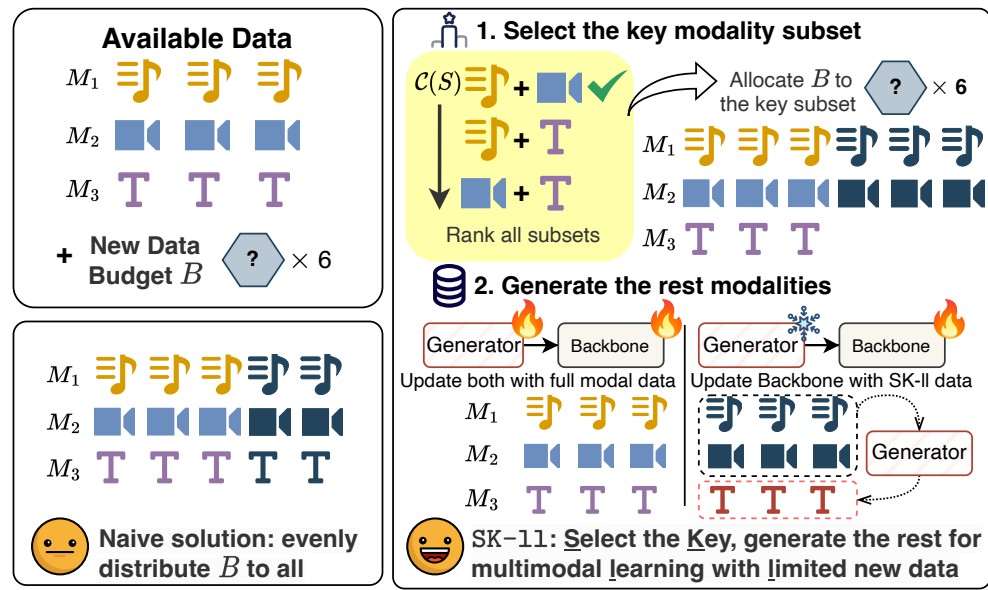

Figure 2: Overview of SK-ll. *Top-left* shows the task setting, where there are some available multimodal data along with a limited new data budget $B$. *Right* illustrates SK-ll framework approaching MML under limited new data with two steps. (1) We first select the key modality subset by evaluating the cooperation indicator $\mathcal{C}(S)$, ranking all possible subsets, and allocating the data budget $B$ to the top subset. (2) We then synthesize missing modalities corresponding to increased sample amounts through generators. In the second step, we start model training by updating both the generator and multimodal backbone with available full modal data, then continue improving the multimodal backbone with new data from SK-ll. *Bottom-left* contrasts with a naive solution that evenly distributes the new data budget to all modalities.

Our goal is to find an optimal allocation strategy defined by $\{\beta_j\}_{j=1}^m$, where each element $\beta_j \in \mathbb{N}_0$ represents the number of additional samples to be collected for modality $M_j$. The allocation must satisfy the following budget constraint: $\sum_{j=1}^m \beta_j \leq B$. The objective of this multimodal data allocation strategy is to maximize the performance of a model trained in the data set that combines existing data $\mathcal{D}$ and newly collected samples $\mathcal{D}'$ according to the allocation strategy.

## 3.2 SELECT THE KEY: MODALITY IMPORTANCE INDICATOR

The introduction of a new modality usually could bring a non-negative effect, as prior work shows that learning with more modalities provably achieves a smaller population risk (Huang et al., 2021). However, the marginal benefit from new modalities may also decrease as more modalities are included. Therefore, being able to proactively select the most useful modalities can help reduce the cost of collecting and maintaining weak modalities, and improve computational efficiency.

There is a line of works that explore how to quantify the contributions of modality and modalities interaction in multimodal learning. (Gat et al., 2021) introduce the perceptual score that assesses the degree to which a model relies on the different subsets of the input features. (Hu et al., 2022) use the Shapley value to evaluate the cross-modal cooperation for the whole dataset. (Wei et al., 2024) further improve the multimodal cooperation at the sample level using the introduced Shapley-based metric. (Wenderoth et al., 2024) introduces InterSHAP that dissects cross-modal interaction from multimodal learning uses the Shapley interaction index.

Our work focuses on measuring the importance of individual modalities and cooperation in a data-limited scenario. We formalize this intuition by defining an indicator of importance for the combination of modality that captures both the marginal contribution of the individual modalities involved and the degree of complementarity between the modalities.

Inspired by the cooperative game theory, the Shapley value (Roth, 1988) $\phi(\cdot, \cdot)$ measures each player's contribution to overall performance via the weighted average of their marginal contributions to all possible coalitions. Multimodal learning is considered a coalition game of $|\mathcal{M}|$ modalities, and the

Shapley value of modality $M_j$ is defined as in Equation 1:

$$\phi_{\mathcal{M}_j}(\mathcal{M}, f_u) := \sum_{S \subseteq \mathcal{M} \setminus \{j\}} \frac{|S|!(|\mathcal{M}| - |S| - 1)!}{|\mathcal{M}|!} (f_u(S \cup \{\mathcal{M}_j\}) - f_u(S)) \tag{1}$$

where $S$ is a modality subset of $\mathcal{M}$, and $j \notin S$. The contribution of each modality and modality subset is measured by the utility function $f_u(\cdot) : 2^M \to \mathbb{R}$.

We quantify the utility of a set of input modalities $S$ by measuring the reduction in the best achievable expected loss when using $S$, compared to that when models have access to no modalities (*i.e.* only a constant prediction). Formally, let $\ell(\cdot, \cdot)$ be a loss function, and $G(S)$ be the set of predictors from observing modality subset $S$, and $G(\emptyset)$ be predicting the same constant $c$.

$$f_u(S) = \min_{c \in G(\emptyset)} \mathbb{E}[\ell(Y, c)] - \min_{g \in G(S)} \mathbb{E}[\ell(Y, g(X))] \tag{2}$$

The first term in equation 2 represents the minimal loss achieved by any constant predictor, providing a baseline level of performance, while the second term is the minimal loss achieved by an optimal predictor using modality subset $S$. This definition captures the intuition that multimodal input, by incorporating diverse sources of information, typically reduces prediction loss. The utility score thus facilitates ranking and selecting the most informative modalities based on their contribution to predictive accuracy. When all modalities are independent, their marginal contributions are distinct; in such cases, greedily selecting the modality with the highest contribution iteratively builds a subset that optimizes the data collection strategy for maximizing model performance.

In such cases, we use a scaled Shapley value to measure each modality's individual contribution, where $Z$ is the total utility gain of the full model, acting as a dataset-level normalization factor (Gat et al., 2021):

$$\widehat{\phi}_{M_j}(\mathcal{M}; f_u) = \frac{1}{Z} \phi_{M_j}(\mathcal{M}; f_u) \tag{3}$$

However, the Shapley value is an additive measure that allocates the total payoff to individual players, effectively marginalizing out specific interaction structures. To explicitly capture how modalities cooperate within a specific subset $S$ (e.g., identifying whether they provide complementary information beyond their individual capabilities), we propose a measure of **Interaction Synergy**.

Specifically, let $S \subseteq \mathcal{M}$ be a subset of modalities. We define the normalized cooperation score $\mathcal{C}(S)$ as the utility gain of the subset adjusted by the sum of independent unimodal utilities:

$$\mathcal{C}(S) = \frac{1}{Z_S} \Big( f_u(S) - \sum_{M_j \in S} f_u(\{M_j\}) \Big), \tag{4}$$

where $f_u(S)$ is the actual utility gain of the combined subset, and $\sum f_u(\{M_j\})$ represents the baseline utility if modalities were assumed to be independent. To ensure this metric is comparable across subsets with varying performance scales, we normalize it by $Z_S = f_u(S)$, which denotes the performance capacity of the subset itself. Intuitively, a positive $\mathcal{C}(S)$ indicates a synergistic effect (the joint performance exceeds the mere sum of individual contributions), whereas a negative score suggests redundancy or interference between modalities.

Table 1 presents the modality importance scores over a few affection computing datasets under limited data availability, using classification accuracy as the utility function. We take the majority-class accuracy as a baseline when certain modalities are absent. All possible permutations are evaluated so that both unimodal contributions and cross-modal interactions are thoroughly assessed.

### 3.3 GENERATE THE REST: MODALITY-INTERFERENCE-AWARE INCOMPLETE MODALITY GENERATION

Informed by SK-ll, we obtain the most useful subset of all modalities $S_{\text{ava}} = \{M_1, M_2, ..., M_p\}$, while the other modalities could not be observed, i.e., $S_{\text{miss}} = M \setminus S_{\text{ava}}$. With available modalities $S_{\text{ava}}$, the subsequent task is to recover missing modalities $S_{\text{miss}}$ conditioned on the available ones for better fusion. The incomplete modality generation process consists of the following three parts: **(1) Shallow feature extraction.** We first extract the shallow features of all available modalities and

project them into the same dimensional space, the input features being $X_{\text{ava}} = \mathbf{X}^{(M_i)}$, $M_i \in S_{\text{ava}}$. Under a fixed missing protocol, various available modality combinations are included, which are presented in Table 1. **(2) Missing Modality Generation.** For classification task, we then use a class-specific flow generation model to recover missing modalities from the perspective of data distribution (Wang et al., 2023a). It is important to note that the feature reconstruction module was canceled due to the lack of ground truth samples for the additionally allocated portion. We also preserve the class-specific design to enhance the discriminability for different classes of samples. Given a sample subset $X_{\text{ava}}$ of class $c$, the normalizing flows $\mathbf{Z}^{(M_i)}$ is thus optimized only by a cross-modal distribution transfer loss defined as:

$$l_{gen} = - \sum_{M_i \in S_{\text{ava}}} \left[ \log p_{Z^{(M_i)}}\left( \mathbf{Z}^{(M_i)} \mid y = c \right) + \log \left| \det \left( \frac{\partial \mathbf{Z}^{(M_i)}}{\partial \mathbf{X}^{(M_i)}} \right) \right| \right] \tag{5}$$

where the first term denotes the log-density of $\mathbf{Z}^{(M_i)}$ on the condition of the label category $c$, and the second term is the log-determinant of normalizing flow model for modality $M_i$. In detail, $Z^{(M_i)} = \mathcal{F}^{(M_i)}(X^{(M_i)}) \sim \mathcal{N}(\mu_c, \Sigma_c)$, $M_i \in S_{\text{ava}}$, where $Z^{(M_i)}$ is the latent state of the modality $M_i$, $\mathcal{F}^{(M_i)}$ is the corresponding forward flow function, and $c$ is the class label of the input sample $X_{\text{ava}}$. The generation objective forces the representations from different modalities to have a similar distribution over the discrete feature space.

Table 1: Accuracy (Acc.) of MuLT on `MOSI` and `MOSEI`, all with 40% training data . The cooperation $\mathcal{C}(\cdot)$ scores over each modality subset $S$, and our proposed indicator to measure the importance of modality in multimodal learning. Rank.: the rank of each subset in $\mathcal{C}(S)$.

| Datasets | $S$ | Acc. | $\mathcal{C}(S)$ | Rank. |
|---|---|---|---|---|
| MOSI | T, A | 75.02 | -0.015 | 2 |
| | T, V | 77.52 | 0.014 | 1 |
| | A, V | 55.63 | -0.066 | 3 |
| MOSEI | T, A | 70.92 | 0.037 | 1 |
| | T, V | 70.35 | 0.029 | 2 |
| | A, V | 52.57 | -0.101 | 3 |

The entire training is implemented in an end-to-end manner, jointly optimizing the model parameters for both multimodal fusion and incomplete modality generation components. We integrate the above losses to reach the full optimization objective as $\mathcal{L} = l_{task} + \alpha \times l_{gen}$, where $l_{\text{task}}$ is the task-specific loss defined as the mean absolute error for the regression task, and $\alpha$ control the importance of the generation loss function and modality conflict loss, respectively.

### 3.4 Overview of Our framework

The overview of our method is illustrated in Figure 2. The complete algorithm for crafting a recipe for dataset in the limited data regime is summarized in Algorithm 1. Algorithm 1 starts with an empty set, and subsequently adds to the current set the new modality that maximizes the marginal gain at each iteration. For each candidate modality $M_i$, we (1) train two models on $S$ and $S \cup M_i$ respectively until training losses converge, and (2) record the difference between two sets of performance metrics, and compute the importance scores; (3) select the top $p$ modalities (where $p$ is the cardinal constraint of modalities that can be selected) based on the ranked importance score, and add the selected modality to $S$ to construct $S_{i+1}$. Normally we end up with more complementary modalities, ensuring that the modality learning utility (i.e., $f_1$ or $f_2$ values) selected in each iteration is a positive value. That is to say, the modality selection process terminates when no modality with a positive effect on the current combination can be found in any iteration. Given the optimal modality combination, we next utilize available data modalities to generate missing modality, to form full modal representations that will be then used in multimodal fusion and prediction.

---

**Algorithm 1** Step-wise Maximization of Modality Selection
___
1: **Data:** Full modality set $M = \{M_1, \ldots, M_{|M|}\}$, fixed sampling budget $B$
2: **Input:** Unimodal contribution $f_1$ (Eq. 3), cross-modal cooperation $f_2$ (Eq. 4), max number of modalities to select $k$
3: **Output:** $\mathcal{S}_k$
4: $S_0 = \emptyset$
5: **for** $i = 0, 1, ..., k - 1$ **do**
6:     **if** $i = 0$ **then**
7:         $M^i = \underset{M_j \in M}{\arg\max}\, f_1(M_j)$
8:     **else**
9:         $M^i = \underset{M_j \in M \setminus \mathcal{S}_i}{\arg\max}\, f_2(\mathcal{S}_i, M_j)$
10:     **end if**
11:     $\mathcal{S}_{i+1} = \mathcal{S}_i \cup \{M^i\}$
12: **end for**

---

## 4 Experiment

### 4.1 Implementation Details

**Backbone Models and Benchmarks.** We evaluate our proposed `SK-ll` across below multimodal learning backbones and application:

• **Affection computing**: We apply SK-ll on Multimodal Transformer (Yu et al., 2019) with Flow-based generation model (Wang et al., 2023a) with the MOSI (Zadeh et al., 2016) and MOSEI (Zadeh et al., 2018) datasets, which provide multimodal data consisting of Text (T), Video (V), and Audio (A), all aligned at the clip level. Specially, the modality Video refers strictly to visual frame features. The MOSI dataset contains 1,284 training, 229 validation, and 686 testing samples, each annotated with sentiment labels. The MOSEI dataset, an extension of MOSI, includes a larger collection with 16,326 training, 1,871 validation, and 4,659 samples, and also provides continuous sentiment intensity labels. Each sample is labeled with a sentiment valence ranging from -3 (strongly negative) to +3 (strongly positive). In addition, we apply SK-ll on a multimodal pre-training CoMM (Dufumier et al., 2025), a multimodal pre-training architecture, with another two affection computing benchmarks UR-Funny (Hasan et al., 2019), MUsTARD (Castro et al., 2019). For evaluation metrics, we use Accuracy2 on all affection computing benchmarks.

• **Healthcare Application**: We apply SK-ll on Flex-MoE (Yun et al., 2024), a MoE-based multimodal model achieving SOTA performance on healthcare benchmarks. We test SK-ll with Alzheimer's Disease Neuroimaging Initiative (ADNI) (Weiner et al., 2010), which involves four key modalities for AD stage prediction, and the Medical Information Mart for Intensive Care IV (MIMIC-IV) dataset (Johnson et al., 2019) spanning across ICD-9 codes, clinical text, and vital values modalities. We only keep the samples with all modalities in MIMIC and ADNI for training. For evaluation, we use F1 as the performance indicator.

**Baselines.** For lower bound of SK-ll, we train a model using a minimal amount of data, representing the base scenario. The baseline approach allocates the data collection budget evenly across all modalities. In the SK-ll experiment, the modality importance indicator selects the key modality combination, and the budget is evenly distributed across this combination. Missing modalities are generated to complete the data. Finally, the **upper bound** scenario allocates a separate, dedicated budget to each modality.

**Training and Evaluation Details.** We conduct four groups of experiments for each dataset, with the lower bound set to 10%, 20%, 30%, and 40%, respectively. Each experiment is repeated five times with different random seeds, and the final results are reported as the mean $\pm$ standard deviation. For all experiments, the data collection budget is fixed at 10%. For example, when the lower bound is set to 10%, the Baseline setting trains the model with 13.33% of the full modalities data, while the upper bound trains with 20% of the full modalities data. For unimodal training, the maximum number of epochs is set to 100, the learning rate is set to 1e-4, the batch size is set to 128, and weight decay is set to 0.005, with the patience for early stopping set to 8. These settings are consistent across all modalities in both the MOSI and CMU-MOSEI datasets. For multimodal training, the maximum number of epochs is set to 100, the learning rate is set to 1e-4, batch size is set to 128, weight decay is set to 0.005, and patience is set to 10. These settings are applied to all possible modality combinations in the affection computing datasets. On the ADNI and MIMIC datasets, we follow the training and evaluation setting as (Yun et al., 2024).

Our experimental setup guarantees fair comparison under the same New Data Budget $B$ by ensuring that the effective increase in complete multi-modal training instances, enabled by the new data budget B, is consistent across our method (SK-ll) and baselines. The budget B defines the total number of new samples introduced. Baselines typically allocate this budget evenly; for example, with $B$=10% and three modalities ($|\mathcal{M}| = 3$), each modality receives about 3.33% new data, resulting in a 3.33% increase in paired triplets. In contrast, SK-ll focus the budget, e.g., selecting two modalities and acquiring one new data for each. To equalize the comparison, for every new sample pair acquired in the selected modalities, SK-ll generates the corresponding features for the unselected modality. This yields a 5% increase in total sample pairs.

### 4.2 SUPERIOR PERFORMANCE OF SK-LL

Based on the results presented in Table 2, serveal key observations can be drawn regarding the performance of our proposed SK-ll compared to the baselines across affection computing (MOSI, MOSEI, UR-Funny, and MUsTARD) and healthcare (MIMIC and ADNI).

**Naively allocating new data budget to all modalities results in suboptimal performance gain and even decrease.** Simply augmenting data by allocating the budget evenly across all modalities (Baseline) is often suboptimal and can even be detrimental. As observed on the MOSI dataset at 10% and 40% available data, the Baseline method performs worse than the configuration using only

Table 2: Performance of `SK-ll` across applied to classification tasks and traditional multimodal model. There are in total 4 groups of experiments, shown in rows with same color (■, ■, ■, ■), with upper and lower adjacent experiments trained on full modality data as upper and lower bound. Best results are **bolden**. "Ava. $\mathcal{D}$" stand for available data ratio. "Gen." stand for whether we generate data embedding for missing modalities. "Baseline" method simply allocate quota to all modalities evenly. For Baseline and `SK-ll` we select data budget $B$ as 10%.

| Ava. $\mathcal{D}$ | Gen. | Method | MulT | | CoMM | | Flex-MoE | |
|---|---|---|---|---|---|---|---|---|
| | | | MOSI | MOSEI | UR-Funny | MUsTARD | ADNI | MIMIC |
| | - | - | $72.01_{\pm 2.54}$ | $69.83_{\pm 1.43}$ | $51.79_{\pm 2.02}$ | $59.07_{\pm 0.32}$ | $30.61_{\pm 7.15}$ | $57.34_{\pm 0.85}$ |
| 10% | ✗ | Baseline | $70.52_{\pm 3.83}$ | $71.94_{\pm 0.33}$ | $51.10_{\pm 2.82}$ | $\mathbf{59.60_{\pm 0.47}}$ | $32.34_{\pm 4.95}$ | $57.70_{\pm 0.61}$ |
| | ✓ | SK-ll | $\mathbf{75.18_{\pm 1.36}}$ | $\mathbf{76.82_{\pm 6.67}}$ | $\mathbf{51.62_{\pm 1.11}}$ | $59.19_{\pm 0.04}$ | $\mathbf{48.87_{\pm 3.65}}$ | $\mathbf{58.41_{\pm 1.05}}$ |
| | - | - | $71.78_{\pm 2.32}$ | $71.07_{\pm 0.67}$ | $50.16_{\pm 2.65}$ | $59.67_{\pm 0.82}$ | $40.90_{\pm 7.24}$ | $59.16_{\pm 0.55}$ |
| 20% | ✗ | Baseline | $76.24_{\pm 0.77}$ | $70.63_{\pm 1.28}$ | $57.72_{\pm 1.11}$ | $59.73_{\pm 0.15}$ | $44.44_{\pm 6.91}$ | $59.78_{\pm 0.75}$ |
| | ✓ | SK-ll | $\mathbf{77.93_{\pm 0.60}}$ | $\mathbf{80.81_{\pm 0.37}}$ | $52.06_{\pm 3.20}$ | $\mathbf{61.61_{\pm 1.02}}$ | $\mathbf{54.06_{\pm 5.49}}$ | $\mathbf{60.13_{\pm 0.81}}$ |
| | - | - | $76.24_{\pm 0.72}$ | $71.29_{\pm 0.65}$ | $52.99_{\pm 0.97}$ | $59.27_{\pm 0.37}$ | $49.69_{\pm 4.55}$ | $60.37_{\pm 0.73}$ |
| 30% | ✗ | Baseline | $76.53_{\pm 1.42}$ | $71.71_{\pm 0.89}$ | $54.28_{\pm 3.15}$ | $61.61_{\pm 0.20}$ | $54.11_{\pm 2.12}$ | $61.52_{\pm 0.83}$ |
| | ✓ | SK-ll | $\mathbf{79.18_{\pm 0.56}}$ | $\mathbf{80.36_{\pm 0.50}}$ | $\mathbf{59.41_{\pm 1.29}}$ | $\mathbf{61.87_{\pm 0.07}}$ | $\mathbf{54.83_{\pm 5.99}}$ | $\mathbf{61.83_{\pm 0.56}}$ |
| | - | - | $77.26_{\pm 0.82}$ | $71.21_{\pm 1.14}$ | $53.86_{\pm 1.52}$ | $60.37_{\pm 0.22}$ | $53.70_{\pm 4.32}$ | $60.48_{\pm 0.82}$ |
| 40% | ✗ | Baseline | $77.11_{\pm 1.70}$ | $71.10_{\pm 0.84}$ | $55.93_{\pm 3.36}$ | $61.51_{\pm 0.29}$ | $54.28_{\pm 2.23}$ | $61.30_{\pm 1.03}$ |
| | ✓ | SK-ll | $\mathbf{79.94_{\pm 0.87}}$ | $\mathbf{80.16_{\pm 0.25}}$ | $\mathbf{57.70_{\pm 2.16}}$ | $\mathbf{62.24_{\pm 0.19}}$ | $\mathbf{61.76_{\pm 0.43}}$ | $\mathbf{62.87_{\pm 0.80}}$ |
| 50% | - | - | $76.82_{\pm 1.93}$ | $71.84_{\pm 0.52}$ | $56.82_{\pm 1.58}$ | $61.40_{\pm 0.48}$ | $63.96_{\pm 3.20}$ | $62.19_{\pm 1.78}$ |

the initial available data, with performance drops. Similarly, on the `UR-Funny` at 10% available data and `MUStARD` at 40% available data, the Baseline yields lower results compared to the results with initial available data. While adding data evenly does provide improvements in some cases (e.g., `ADNI`), it underscore that naive data augmentation does not guarantee performance gains, particularly in very limited data regimes if the added data modalities does not effectively contribute to multimodal learning.

**`SK-ll` surpasses Baseline with significant margins.** Our proposed method `SK-ll`, which first determines key modalities for budget allocation and generates missing data, demonstrates substantial performance improvements across the all benchmarks. It consistently outperforms the initial available data setting and, in most cases, surpasses the Baseline approach, often by significant margins. ❶ `SK-ll` generalizes across tasks and multimodal backbones: The effectiveness of `SK-ll` is not confined to a specific problem type or model architecture, showing robust performance across diverse settings. On affective computing we experiment with both fusion MML model (MulT) and self-supervised learning MML model (CoMM). On the `MOSI` and `MOSEI` datasets using MulT, `SK-ll` consistently yields significant gains over Baselines. While on healthcare tasks (`ADNI` and `MIMIC`) with Mixture-of-Experts style MML model Flex-MoE, `SK-ll` also consistently delivers strong results. On more abudant benmark `ADNI`, `SK-ll` even achieve a 6.726% average increase over Baseline. These consistent improvment across different task domains and underlying multimodal architectures highlights the broad applicability of our selective budget allocation strategy. ❷ `SK-ll` generalize across data availability ratios: Furthermore, the advantages of `SK-ll` hold across the spectrum of initial data availability ratios tested from 10% to 40%. On relative low available data ratios, `SK-ll` provides substantial benefits even when starting with minimal data. The improvement on `MOSEI` at the 10% lower bound (76.82 vs 71.94 Baseline) exemplifies this. Moreover, `SK-ll` continues to outperform as initial data increases, even with a relatively larger initial dataset (40% lower bound), `SK-ll` maintains its edge, as seen on `MIMIC` (62.87 vs 61.30 Baseline) and `MOSEI` (80.16 vs 71.10 Baseline). This underscores the robustness of our approach. These findings demonstrates effective budget utilization regardless of the initial data scale. The overall trend strongly supports that `SK-ll` offers a robust, generalizable, and data-efficient approach to enhancing multimodal model performance through strategic data collection under limited data scenario.

### 4.3 EXTRA STUDIES

**`SK-ll` Performs Consistently Under Varied New Data Budgets** In our primary experiments, we fixed the new data budget to 10% across varying availability levels. To examine the generalizability of `SK-ll` across different data budget scenarios, we conduct additional experiments using three different budget ratios (10%, 30%, and 50%), representing multimodal learning contexts ranging from low to high resource availability. The results presented in Table 3 demonstrate consistent and

Table 3: Performance of `SK-ll` across applied to multimodal models trained with 10%, 30%, 50% data, with varied data increment 10%, 30%, 50% . The dataset is `MOSI` and metrics is Acc. New data collection quota is set to $i \times \frac{1}{|M|}$ per increment, where $M$ is the number of modalities.

| Ava. $\mathcal{D}$ | +10% Data | | +30% Data | | +50% Data | |
|---|---|---|---|---|---|---|
| | Baseline | `SK-ll` | Baseline | `SK-ll` | Baseline | `SK-ll` |
| 10% | $70.52_{\pm 3.83}$ | $\mathbf{75.18_{\pm 1.36}}$ | $71.78_{\pm 2.32}$ | $\mathbf{75.83_{\pm 1.42}}$ | $73.61_{\pm 4.84}$ | $\mathbf{76.12_{\pm 0.71}}$ |
| 30% | $76.53_{\pm 1.42}$ | $\mathbf{79.18_{\pm 0.56}}$ | $77.26_{\pm 0.82}$ | $\mathbf{77.67_{\pm 0.95}}$ | $77.63_{\pm 1.58}$ | $\mathbf{79.59_{\pm 0.39}}$ |
| 50% | $77.93_{\pm 0.99}$ | $\mathbf{79.24_{\pm 0.70}}$ | $78.68_{\pm 2.56}$ | $\mathbf{79.49_{\pm 0.68}}$ | $79.13_{\pm 1.94}$ | $\mathbf{80.10_{\pm 0.44}}$ |

significant performance improvements by `SK-ll` across all tested budget settings, highlighting its robustness and flexibility in managing data resource constraints.

Table 4: Ablation study on the Budget Allocation Strategy. We fix the modality subset from the "Selection" step and vary the budget ratio assigned to the identified "Primary Modality." The baseline is a uniform split (50% for 2 modalities). The dataset is `MOSI` and metrics is Acc2.

| Ava. $\mathcal{D}$ | Modality Subset | | Ratio for Primary Modality | | | |
|---|---|---|---|---|---|---|
| | Selected | Primary | 50% | 60% | 70% | 80% |
| 10% | $T, A$ | $T$ | $\mathbf{75.18_{\pm 1.36}}$ | $74.17_{\pm 0.92}$ | $73.89_{\pm 1.70}$ | $74.05_{\pm 1.09}$ |
| 30% | $T, V$ | $T$ | $\mathbf{79.18_{\pm 0.56}}$ | $78.20_{\pm 1.36}$ | $77.86_{\pm 0.61}$ | $77.67_{\pm 0.60}$ |
| 50% | $T, V$ | $T$ | $\mathbf{79.24_{\pm 0.70}}$ | $78.24_{\pm 0.70}$ | $77.90_{\pm 0.58}$ | $77.36_{\pm 0.76}$ |

**Balanced Data Allocation Strategy Improves Efficiency and Stability** We investigate the effectiveness of different budget allocation strategies for newly collected data. Specifically, we compare a weighted allocation strategy, where modalities within a subset are ranked according to their single-modal Shapley values, prioritizing modalities with higher contributions (denoted as the "Primary Modality") against a balanced allocation. Results summarized in Table 4 indicate that performance differences among various allocation strategies are marginal, with excessively emphasizing the primary modality potentially harming overall performance. Consequently, for simplicity and pipeline efficiency, we uniformly distribute the data budget among modalities, as applied in our primary results.

**Evaluating the Modality Importance Indicator.** First, to evaluate the effectiveness of our proposed modality importance indicator, we conduct a case study using the `MOSI` dataset. We separately allocate data collection budgets to all possible modality combinations and compare the modality importance ranks from the indicator with the actual experimental results. With a lower bound set at 20%, we rank the combinations using Algorithm 1, which orders them as $\{T, A\}$, $\{T, V\}$, and $\{A, V\}$. The experimental results align with the indicator's ranking, with $\{T, A\}$ yielding the best performance and $\{A, V\}$ the poorest. Detailed results can be found in Appendix A.1. Second, to verify the reliability of the indicator, we test its stability under varying amounts of initial available data. We compute the cooperation score $\mathcal{C}(S)$ and the actual downstream task accuracy for all modality pair combinations on the MOSI dataset, with available data ratios ranging from 10% to 40%. As presented in Table 5, the optimal modality subset for data collection can shift as more data becomes available—for instance, changing from $\{T, A\}$ at 10% and 20% data to $\{T, V\}$ at 30% and 40% . Our indicator $\mathcal{C}(S)$ correctly tracks this shift, consistently identifying the best-performing combination at each stage. This result confirms that our indicator is robust and reliably adapts to different data-limited scenarios, providing a solid foundation for our selection strategy.

**Fairness Audit in Healthcare Application** To evaluate the fairness and potential biases of our method in sensitive domains, we conducted a fairness audit on the `ADNI` healthcare dataset. We disaggregated the model's performance by gender and age subgroups, comparing our `SK-ll` framework against the uniform-allocation baseline and no-generation models under a 40% available data and 10% new data budget setting. As shown in Table 6, `SK-ll` mitigates the age-related accuracy gap observed in the baseline (0.067 vs. 0.094) while maintaining a comparable gender gap. This analysis suggests that our strategic data selection and generation process not only enhances overall accuracy but also promotes more equitable performance across different patient populations.

Table 5: The stability of our modality importance indicator $\mathcal{C}(S)$ on MOSI. The indicator's ranking aligns with the actual subset performance (Acc2) as the amount of available data changes. Best performing subset at each data ratio is bolded.

| Ava. $\mathcal{D}$ | Accuracy (Acc) | | | Indicator $\mathcal{C}(S)$ Value | | |
|---|---|---|---|---|---|---|
| | T+A | T+V | A+V | T+A | T+V | A+V |
| 10% | **71.05**$_{\pm 1.16}$ | 68.48$_{\pm 2.26}$ | 48.57$_{\pm 4.63}$ | **0.120** | 0.035 | 0.078 |
| 20% | **76.33**$_{\pm 0.48}$ | 73.00$_{\pm 1.16}$ | 52.65$_{\pm 2.05}$ | **0.092** | 0.035 | -0.010 |
| 30% | 75.54$_{\pm 1.44}$ | **76.70**$_{\pm 1.39}$ | 55.02$_{\pm 1.50}$ | -0.008 | **0.041** | 0.006 |
| 40% | 75.02$_{\pm 1.31}$ | **77.52**$_{\pm 0.64}$ | 55.63$_{\pm 1.29}$ | -0.015 | **0.014** | -0.066 |

Table 6: Model performance by subgroups on the ADNI dataset.

| Model and Ava. $\mathcal{D}$ | Male Acc. | Female Acc. | Gender Gap (Abs.) | Young Acc. | Old Acc. | Age Gap (Abs.) |
|---|---|---|---|---|---|---|
| No-Gen (40%) | 0.489 | 0.485 | 0.004 | 0.457 | 0.492 | 0.035 |
| Baseline (40%+10%) | 0.552 | 0.549 | 0.003 | 0.457 | 0.551 | 0.094 |
| **SK-ll (40%+10%)** | **0.631** | **0.655** | 0.024 | **0.571** | **0.638** | **0.067** |
| No-Gen (50%) | 0.659 | 0.645 | 0.014 | 0.714 | 0.647 | 0.067 |

## 5 CONCLUSION

In this paper, we address the practical challenge of enhancing multimodal learning in data-limited scenarios where new data acquisition is constrained by a finite budget. We propose SK-ll, a novel framework designed to strategically maximize the performance gain from this budget. Our framework operates in two stages: it first utilizes a modality importance indicator, grounded in Shapley values, to identify the most synergistic subset of modalities for data collection . Then, it allocates the entire budget to this key subset and employs a generative model to synthesize the missing modalities, creating complete data for robust training on downstream tasks.

Our extensive experiments across diverse affective computing and healthcare benchmarks demonstrate that SK-ll consistently and significantly outperforms naive, uniform budget allocation strategies. We show that this strategic selection leads to more stable training dynamics and that the approach is robust across various initial data availability ratios and generalizes across different multimodal backbones. Ultimately, SK-ll presents a principled and data-efficient paradigm for resource-constrained MML, shifting the focus from retrospectively handling missing data to prospectively optimizing data collection.

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

# A APPENDIX

## A.1 EXTRA EXPERIMENTS

Table 7: Results of different modality combination for generation.

| Combination | Acc2 | F1 Score | Acc7 |
|---|---|---|---|
| T+A | **77.29±0.78** | **77.33±0.72** | **29.26±0.60** |
| T+V | 75.88±2.00 | 75.91±2.06 | 28.69±1.92 |
| A+V | 63.14±11.72 | 59.11±15.78 | 20.44±6.89 |

**Evaluating Modality Importance Indicator** As shown in Table 7, we allocate extra data collection budget to all possible modality combinations and then report corresponding results. When setting lower bound to 10%, our modality importance indicator orders the modality combinations as T+A, followed by T+V, then A+V. The results shown in Table 7 is consistent with the order obtained by indicator, proving the effectiveness of our proposed modality importance indicator.

**Evaluating `SK-ll` under Different Testing Conditions.** To assess the robustness of `SK-ll`, we take `MOSI` as an example, evaluating it under various testing conditions using combinations of $T$, $A$, and $V$ modalities: $\{T, A\}$, $\{T, V\}$, $\{A, V\}$, and $\{T, A, V\}$. For instance, the T+A condition uses only the Text and Audio modalities during testing. All results are based on the best models validated with the full validation set. As shown in Table 8, the $\{T, A\}$ and $\{T, V\}$ conditions perform similarly to the full testing set $\{T, A, V\}$, demonstrating the model's robustness. In contrast, $\{A, V\}$ yields lower results, which aligns with the indicator's findings that $\{A, V\}$ is less suitable for budget allocation.

Table 8: Ablation results under different testing conditions across different available data (Ava. $\mathcal{D}$) on `MOSI`.

| Ava. $\mathcal{D}$ | Testing Conditions | | | |
|---|---|---|---|---|
| | $\{T, A\}$ | $\{T, V\}$ | $\{A, V\}$ | $\{T, A, V\}$ |
| 0.1 | 72.26 | 73.72 | 44.34 | **75.65** |
| 0.2 | 73.96 | 76.44 | 47.82 | **77.95** |
| 0.3 | 72.77 | **79.19** | 36.84 | 79.08 |
| 0.4 | 74.26 | 72.81 | 44.84 | **80.07** |

Table 9: Comparison of different generative models on MOSI (20% available data + 10% new data budget).

| Generator Model | Accuracy (Acc2) | Avg. Training Time per Epoch |
|---|---|---|
| Normalizing Flow (Ours) | **77.93±0.60** | **~3s** |
| Diffusion | 75.49±1.28 | ~15s |

**Ablation Study on Generative Models** To evaluate our choice of the generative model, we conduct a comparative study between our normalizing flow-based approach and a diffusion-based alternative. The experiment was performed on the MOSI dataset, using 20% available data with an additional 10% new data budget allocated according to `SK-ll`. The results, summarized in Table 9, demonstrate that the normalizing flow model not only achieves superior accuracy but also exhibits significantly higher computational efficiency, with a much shorter training time per epoch. Given that our framework prioritizes both budget and resource efficiency, this outcome validates that the normalizing flow model offers the best trade-off between predictive performance and computational cost for our task.

**Computational Cost and Scalability Analysis** A practical consideration for our `SK-ll` framework is the up-front computational cost required to calculate the modality importance indicator, as this process involves training or evaluating a model for each modality subset. To address this concern, we provide a concrete benchmark of this one-time cost. As shown in Table 10, we measured the total wall-clock time for the indicator computation on a single NVIDIA A6000 GPU. The results show that this one-time cost is moderate and practical for model deployment, especially considering that we utilize early stopping to reduce actual training time. Furthermore, we demonstrate that this up-front cost can be drastically reduced without compromising the outcome of modality selection. We conducted an experiment on the MOSI dataset where we computed the indicator using just 100 random samples and compared the resulting subset ranking to that derived from all available data. The results in Table 11 show that while the absolute $\mathcal{C}(S)$ scores differ, the relative ranking of the modality subsets is perfectly preserved. This finding confirms that the indicator can be reliably estimated from a small data sample, significantly enhancing the efficiency and practical applicability of `SK-ll` framework.

Table 10: One-Time Indicator Computation and Model Training Time. We report the total wall-clock time (in minutes) for the modality selection process on a single NVIDIA A6000 GPU across different datasets and data availability ratios.

| Dataset | Modalities | Available Data | `SK-ll` Indicator Computation | `SK-ll` Model Training | Baseline Model Training |
|---------|-----------|----------------|-------------------------------|------------------------|-------------------------|
| MOSI | 3 | 10% | 8.13 | ∼2.5 | ∼1.8 |
| | | 50% | 13.99 | ∼3.5 | ∼2.5 |
| MOSEI | 3 | 10% | 12.07 | ∼3.8 | ∼3.0 |
| | | 50% | 22.2 | ∼6.5 | ∼5.0 |
| ADNI | 4 | 10% | 73 | ∼12 | ∼9 |
| | | 50% | 147 | ∼25 | ∼18 |
| MIMIC | 3 | 10% | 39 | ∼8 | ∼6 |
| | | 50% | 137 | ∼28 | ∼22 |

Table 11: Indicator Scores C(S) on MOSI computed from all available data versus 5 non-overlap small random subset of 100 samples. The relative ranking of subsets (T,V > T,A > A,V) is preserved, demonstrating the feasibility of using a small sample for efficient estimation.

| Modality Subset | Acc. with All Data | C(S) with All Data | C(S) with 100 Samples |
|-----------------|--------------------|--------------------|-----------------------|
| T, V | 77.52 | 0.014 | $0.121 \pm 0.008$ |
| T, A | 75.02 | -0.015 | $0.062 \pm 0.007$ |
| A, V | 55.63 | -0.066 | $-0.094 \pm 0.005$ |

## A.2 THE USAGE OF LLM

GPT-5 was employed for language refinement purposes only. Its application was confined to: (1) proofreading for typographical errors, and (2) correcting grammatical mistakes. The AI tool had no role in the formulation of research ideas, experimental design, data analysis, or the generation of any scientific content.

## A.3 EXTRA IMPLEMENTATION DETAILS

**Generation Model Details** For the MulT and CoMM backbones, we employ a class-conditional Normalizing Flow to synthesize missing modalities. The architecture comprises a shallow feature extractor (kernel size 5), a cross-modal distribution transfer module consisting of 32 invertible layers, and a reconstruction decoder utilizing 20 Residual Channel Attention Blocks. To accommodate varying dataset complexities, the hidden dimensions are set to 64 for CMU-MOSI and 128 for CMU-MOSEI. Conversely, for Flex-MoE, we utilize its learnable missing modality bank rather than a separate generator network. This framework is configured with a hidden dimension of 128 across tasks; specifically, on the ADNI dataset, it employs 16 experts with a Top-4 gating strategy, while on MIMIC-IV, it scales to 32 experts with a Top-3 gating strategy to effectively model diverse modality combinations.

## A.4 VISUALIZATION OF GENERATED DATA

To qualitatively evaluate feature generation quality, we visualize the distributions of real and generated data using t-SNE (van der Maaten & Hinton, 2008). We randomly select 100 testing samples from `MOSI`, projecting their multimodal features onto a 2D plane. As illustrated in Figure 3, the generated features reside in the same latent manifold as the authentic data, preserving modality-specific structures. This confirms that `SK-ll` effectively synthesizes realistic representations without additional data collection. Furthermore, comparing the visualization across different data settings reveals that the feature clusters become more distinct and structured with increased availability of initial training data (from 10% to 50%). This suggests that `SK-ll` effectively leverages richer data contexts to refine the separability and quality of the synthetic features.

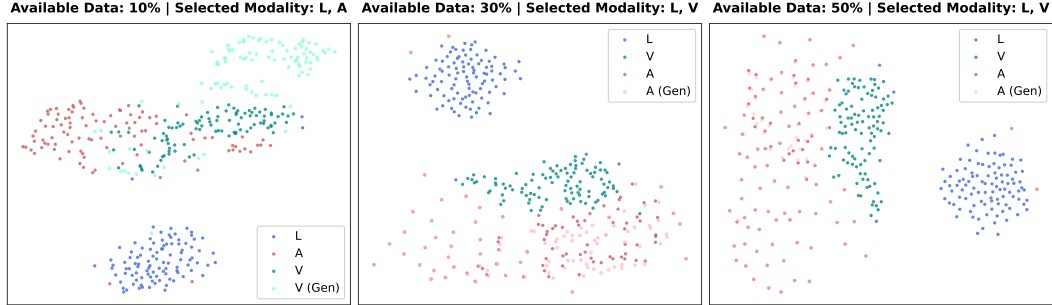

Figure 3: Visualization of real and synthetically generated multimodal features (`MOSI`), at different availability levels (10%, 30%, and 50%) with an additional 10% data budget. Modality abbreviations: Text ($T$), Video ($V$), Audio ($A$).

