# OpenReview forum: "Select the Key, Then Generate the Rest: Improving Multi-Modal Learning with Limited Data Budget"
_ICLR.cc/2026/Conference — Submitted to ICLR 2026_

### Official Review · Reviewer_RKhA · 2025-10-27

**Soundness:** 2
**Presentation:** 1
**Contribution:** 2
**Rating:** 2
**Confidence:** 4

**Summary:**

The paper proposes Select the Key modality, then generate the rest to enable learning with limited data (SK-ll) as a framework of efficient multimodal learning under a constrained data collection budget. The pipeline first selects the most useful modality using a modality importance indicator that considers both the modality's marginal contribution and its cooperation with other modalities. Then the pipeline has another class-specific flow generation model to generate the missing ones. The proposed pipeline is evaluated on several real-world affection computing tasks and a healthcare downstream task to show its effectiveness over the naive allocation baseline.

**Strengths:**

- The evaluation study is relatively complete, which evaluates the proposed method on several affect computing tasks, and the experiment setups are well-documented;
- The paper includes a detailed analysis and discussion following the experiment results to highlight the efficiency and stability of the method;

**Weaknesses:**

- The sentence "multimodal models with only a subset of modalities available for new data could reach and even surpass models continuously trained with full modalities" in the abstract seems to contradict the claim that "The introduction of a new modality usually could bring a non-negative effect" (beginning of section 3.2) and later, "This definition captures the intuition that multimodal input, by
incorporating diverse sources of information, typically reduces prediction loss" (line 228-229), as well as several existing theoretical findings such as [1]. The paper does not provide a solid evidence / experiment to support this claim in the abstract, except for 2 small plots in Figure 1, where there is hardly any outperformance of model trained with subset of modalities over full-modality training (the only noticeable case is the video+text on MOSI, while the margin shown in the plot is hard to interpret and can come from experiment noises);
- Modality selection is the most relevant research question in multimodal learning with constrained resources, which is also listed as one of the primary contributions of the paper. But the paper (1) **misses the relevant discussion in related work**, and (2) **misses comparisons to several existing methods, especially [1], which the reviewer finds highly similar formulation** that uses Shapley value to characterize marginal-contribution-based feature importance scores. The paper should **clearly differentiate itself from those existing methods**, and highlight any additional contribution on top of the overlapping parts;
- **The comparisons between the proposed method and the naive allocation baseline are not fair**. Despite of having the same budget allocating to collect data, the proposed method also requires additional compute to generate the missing modalities, which naturally brings it advantage in terms of the amount of data / compute that the naive allocation baseline does not have. A more proper comparison should be comparing the proposed method with the naive allocation baseline that leverages comparable compute in (at least) training (if generation is not applicable);
- The writing and presentation can be **much further improved** and the reviewer suggests a more careful proofread of the submission. The following listed items are not considered as typos but rather **either significant formatting issues or have caused confusion for understanding**:
1. Incorrect citation format, please different the use of \cite{...} and \citep{...};
2. Misleading use of "Left"/"Right" in Figure 1 caption, which should be "Top"/"Bottom";
3. The three modalities for MOSI and MOSEI (Figure 1) should be Text, Image (instead of Video), and Audio, as Video naturally contains audio and its text transcript (if not explicit). Labeling it as Video causes the confusion whether this modality already contains information of the other two;
3. Inconsistent interchanged notation of $M$ and $\mathcal{M}$: e.g. the domain of a function $f_u$ should be $2^\mathcal{M}$ instead of $2^M$ in the definition of Equation 1, and mixed $M$, $\mathcal{M}$, mixed $\phi$, $\hat{\phi}$ in the definition of Equation 4;
4. Texts starting from line 277 have a smaller font size compared to the rest of main text (before 277);

[1] He, Y., Cheng, R., Balasubramaniam, G., Tsai, Y.-H. H., & Zhao, H. (2024). Efficient modality selection in multimodal learning. Journal of Machine Learning Research, 25(47), 1–39.

**Questions:**

- Weakness 1: what is the concrete (statistically significant) evidence that supports the claim "multimodal models with only a subset of modalities available for new data could reach and even surpass models continuously trained with full modalities"? And why is it not a contradiction to several existing studies that show the non-negative information advantage from a newly introduced modality?

---

> ### Author Response · Authors · 2025-11-28
>
> We thank the reviewer RKhA for their detailed response. We value the pointer to relevant studies and the opportunity to clarify our problem setting:
>
> **1. Response to Concerns on "Subset vs. Full Modalities" Contradiction**
>
> We respectfully clarify that the apparent contradiction stems from the gap between Information-Theoretic Upper Bounds (ideal scenario) and Optimization Dynamics (practical reality). Below, we provide the theoretical and practical justification, literature support, and concrete statistical evidence from our experiments to resolve this concern.
>
> - Theoretical works such as [1] positing that adding a modality strictly reduces the Bayes error rate **rely on strong, idealized assumptions**, such as infinite training data and perfect optimization capacity. These theories represent an information-theoretic upper bound that implies potential gain, but they do not account for the optimization difficulties in training deep neural networks.
>
> - In contrast, "Less is More" is a common finding in practical Deep Learning. Current multimodal backbones are imperfect, as jointly optimizing heterogeneous modalities often leads to modality competition [2-6], gradient conflict [7], and overfitting to noise [8-9]. When a new modality is noisy or harder to learn, it distracts the optimizer, causing the multimodal model to perform worse than a well-tuned subset.
>
> - Our experiments provide statistically significant evidence of this phenomenon with results obtained from 5-seed:
>     - Evidence of Modality Interference: For example, In Table 2 (MOSI, 10% data), the "Baseline" (augmenting all modalities) achieves 70.52%, which is lower than the "10% Available Data" lower bound (72.01%). This proves that blindly adding data to all modalities hurts performance compared to the initial subset.
>     - Critical Impact of Modality Selection: As demonstrated in Table 5, the choice of modalities dictates performance. On the MOSI dataset, the optimal subset $\{T, V\}$ achieves a high accuracy of 77.52%, whereas the noisy subset $\{A, V\}$ drastically underperforms at 55.63%. This performance gap confirms that identifying the "Key" subset is essential to boost multimodal learning performance.
>
>
> **2. Response to Relevant Work**
>
> We have included a citation and discussion of [10] in our revised Related Work section. While we acknowledge that both methods utilize Shapley values as the mathematical tool, we clarify that our contributions diverge fundamentally in Problem Scope and Methodological Paradigm:
>
> - Problem Scope: While both methods utilize Shapley values for selection, [10] focus on Inference Efficiency (selecting modalities to reduce computational FLOPs and discarding the rest). SK-ll focuses on Data Acquisition Budget.  We aim to minimize the high economic or physical costs of collecting data (e.g., expensive PET scans in ADNI) before new data are present. These are distinct optimization problems with different constraints.
>
> - Methodological Paradigm: Another critical technical difference is that [10] prunes the unselected modalities entirely. In contrast, SK-ll generates the unselected modalities. This "Select then Generate" paradigm allows our backbone to maintain a full-modality representation space, which is crucial for maximizing the performance of the selected "key" modalities, a capability that standard pruning methods do not possess.
>
> **3. Response for comparison fairness**
>
> We respectfully clarify that our experimental setup is grounded in the reality challenges, rather than purely computational equality.
>
> - Clarification of "Budget". In our problem formulation (Section 3), "Budget" ($B$) strictly refers to the Data Acquisition Cost (e.g., the cost of sensors, human labeling effort, or collecting hard-to-obtain samples), not the computational FLOPs.
>     - Domain Context: In our target domains (healthcare and affective computing), the cost of acquiring a new modality sample is orders of magnitude higher than the cost of training a generative model.
>     - Therefore, the most appropriate comparison for this domain is between methods that consume the same Acquisition Budget. SK-ll and the Baseline are directly comparable because they both "spend" the exact same amount of limited new data samples ($N$) from the environment. SK-ll trades a marginal increase in offline training compute for a significant gain in data efficiency.
>
> - To further address the concern about "data/compute advantage," we also compare SK-ll to the "Upper Bound" (Full Modalities) results in Table 2. The Upper Bound model utilizes 100% Real Data (which is inherently superior to synthetic data). Our results show that SK-ll using only a fraction of the data budget + generation performs comparably to this Upper Bound. This confirms that SK-ll is not merely benefiting from "extra compute," but is fundamentally more data-efficient by identifying and amplifying the key modalities.

---

> > ### Author Response · Authors · 2025-11-28
> >
> > **4. Presentation and Typos.**
> >
> > We are thankful for the careful review regarding formatting.
> >
> > - Video vs. Image: We labeled it "Video" because the raw data is video, but we agree that since we extract frame-level features, "Visual/Image" is more precise. We have clarified this in Section 4.1 Implementation Details. We state that the "Video" modality in our paper refers to the visual features extracted from frames.
> >
> > - Formatting: We have reviewed the citation style, the Figure 1 caption ("Top/Bottom"), and the font size inconsistency in the final version.
> >
> > [1] What makes multi-modal learning better than single (provably).
> >
> > [2] What makes training multi-modal classification networks hard?
> >
> > [3] Balanced multimodal learning via on-the-fly gradient modulation
> >
> > [4] Pmr: Prototypical modal rebalance for multimodal learning
> >
> > [5] On uni-modal feature learning in supervised multi-modal learning
> >
> > [6] Multimodal patient representation learning with missing modalities and labels
> >
> > [7] Mitigating modality collapse in multimodal vaes via impartial optimization
> >
> > [8] Multimodal sentiment analysis with word-level fusion and reinforcement learning
> >
> > [9] Are multimodal transformers robust to missing modality?
> >
> > [10] Efficient modality selection in multimodal learning.

---

### Official Review · Reviewer_wGxh · 2025-10-31

**Soundness:** 3
**Presentation:** 3
**Contribution:** 2
**Rating:** 6
**Confidence:** 4

**Summary:**

This paper addresses the practical challenge of multimodal learning (MML) under constrained data collection budgets. To solve this problem, the proposed method, SK-ll, first identifies the most informative modality subset using a modality importance indicator based on marginal contributions and cross-modal interactions, and then generates the remaining modalities via learned embeddings. The framework is validated on affective computing (MOSI, MOSEI) and healthcare datasets (ADNI, MIMIC), achieving average performance improvements of 2–7% over baselines. The results suggest that finding the keys and then generating the rest can yield comparable or even superior performance to models trained on full multimodal data.

**Strengths:**

- This paper studies an important problem that modalities in multimodal learning are not equal, and their importance could vary. Thus, wisely selecting the important ones for training could help the overall performance.
- The two-step method, select and generate, is quite straightforward and intuitive.
- The performance improvement has been extensively validated over quantitative experiments and qualitative studies.

**Weaknesses:**

- The proposed method requires a search over all possible combinations, and it computes Shapley values and cooperation scores over all possible modality subsets, which is not computationally friendly.
- The method computes the utility function, which is dependent on the expected reduction in prediction loss. Therefore, it requires multiple rounds of model training, thus could lead to cost increase and being misled by training randomness.
- The method assumes that modalities are independent of each other, which could be invalid in many real-world scenarios. Many modalities could be correlated to each other; thus, the assumptions could not hold, and it needs more justification is needed to support the proposed method.
- The proposed SK-LL is not end-to-end trainable, which requires multistep training. Moreover, the ablation study seems to be missing in understanding which step is more important in contributing to the overall performance.
- Missing several references:
  - Wei et al., Mmpareto: Boosting multimodal learning with innocent unimodal assistance, in ICML 2024.
  - Yang et al., Facilitating multimodal classification via dynamically learning modality gap, in NeurIPS 2024.
  - Huang et al., Towards Out-of-Modal Generalization without Instance-level Modal Correspondence, in ICLR 2025.

**Questions:**

Please address the questions in Weaknesses.

---

> ### Author Response · Authors · 2025-11-28
>
> We thank the reviewer for recognizing the practical value of our work and for highlighting our extensive validation on affective and healthcare datasets. We appreciate the constructive feedback, which we address below.
>
> **1. Response to Computational complexity and Training costs.**
>
> We acknowledge the computational overhead of the search process but demonstrate that it is both manageable and justified:
>
> - Multimodal tasks typically involve a small cardinality of modalities ($3 \le M \le 6$). This keeps the combinatorial search space computationally feasible.
>
> - The indicator calculation is a one-time up-front cost. This investment reduces the expensive, recurring costs of real-world data acquisition (e.g., medical sensors) required during deployment.
>
> - As shown in Table 11, we demonstrate that the indicator can be estimated using few samples. This preserves the relative ranking of modality importance perfectly while drastically reducing computational time.
>
>
> **2. Response to Validity of the Independence Assumption.**
>
> We respectfully clarify that our method does not assume modalities are independent. Our approach follows a rigorous line of research that utilizes Shapley-based metrics specifically to dissect multimodal interactions and correlations [1-3].
>
> - As defined in Eq. 1, the Shapley value is calculated based on the marginal contribution of a modality when added to various subsets. If Modality A is highly correlated with B, adding A to a subset containing B yields a minimal performance gain. The metric inherently captures this correlation by penalizing redundancy.
>
> - We then define the Cooperation Score in Eq. 4 to quantify the non-additive interactions between modalities. A positive Cooperation Score indicates synergy (where modalities correct each other), while a negative score indicates redundancy.
>
> [1] Perceptual score: What data modalities does your model perceive?
>
> [2] Shape: An unified approach to evaluate the contribution and cooperation of individual modalities
>
> [3] Measuring Cross-Modal Interactions in Multimodal Models
>
> **3. Response to Additional References.**
>
> We thank the reviewer for identifying these recent relevant works. We have incorporated them into the "Related Work" section to better position SK-ll within the current literature. [4-5] focus on improving joint optimization and feature alignment (handling gaps/conflicts) on already collected data. Our SK-ll focuses on the data acquisition stage by strategically deciding what to collect under budget constraints. [6] aims for generalization without correspondence, SK-ll aims to construct correspondence via generation to maximize the utility of the selected "key" modalities.
>
> [4] Wei et al., Mmpareto: Boosting multimodal learning with innocent unimodal assistance, in ICML 2024.
>
> [5] Yang et al., Facilitating multimodal classification via dynamically learning modality gap, in NeurIPS 2024.
>
> [6] Huang et al., Towards Out-of-Modal Generalization without Instance-level Modal Correspondence, in ICLR 2025.
>
>
> **4. Response to End-to-End training and Ablation Studies.**
>
> We thank the reviewer for raising the discussion on the training pipeline and component contributions. We clarify our design choices and the roles of each step below:
>
> - We adopt a multi-stage framework rather than an end-to-end one for two key reasons:
>     -  (1) Generalizability: The "Select" step is designed to be model-agnostic. By decoupling it, our framework can be seamlessly applied to any downstream multimodal backbone (e.g., MulT, CoMM, Flex-MoE) without needing to redesign an end-to-end architecture for each specific model.
>     - (2) Cost Control: The selection process is a one-time, offline cost. Once the key modalities are identified, they guide the physical data collection process. An end-to-end trainable selection would imply updating the selection policy dynamically during training, which contradicts the practical constraint where data collection budgets are often fixed/planned in advance.
>
> - We respectfully point out that our experimental design explicitly disentangles the contributions of the "Select" and "Generate" steps through our baselines and analyses in Table 2:
>     - The Baseline utilizes the same budget and generation mechanism but lacks the "Key Selection" (it allocates evenly/naively). The performance gap quantifies the specific contribution of the Selection strategy. If we remove the "Select" step, the method degrades to this Baseline
>     - If we remove the "Generate" step, the method degrades to training on the initial limited subset, corresponding to the lower bound setting. Therefore, our experiments already cover the full spectrum.

---

### Official Review · Reviewer_6hZs · 2025-10-31

**Soundness:** 4
**Presentation:** 2
**Contribution:** 4
**Rating:** 8
**Confidence:** 3

**Summary:**

In this work, the authors formulates an interesting problem: given some available data in a multimodal setting and some budget to collect additional data from any subset of modalities, how to best select modalities and utilize the budget/additional data to best improve model performance? The proposed method first utilizes Shapley values to determine best modality combinations, then uses a flow-based method to generate the remaining modalities for the newly collected data. The proposed method was evaluated over 6 datasets spanning affective computing and healthcare, and the experiment results demonstrates the proposed method's effectiveness over all of them. Additional analysis was conducted over data allocation balance, accuracy of modality improvement indication, and fairness in healthcare datasets.

**Strengths:**

1. The formulated problem is very interesting and important: when given a budget to collect additional data from any subset of modalities in a multimodal setting, what is the optimal way to collect them that would benefit the overall model performance the most? Tackling this problem can have an impact on many real world situations, where data collection can incur a cost from a limited total budget.

2. The authors proposed a novel method to cleverly utilize Shapley values to determine the most useful modality combinations that provides the most complementary contribution to overall performance. This approach may also be useful for better understanding modality contribution and balance within various multimodal classification tasks.

3. The method was not only evaluated on affective computing with TVA modalities, but also on healthcare datasets with very different data modalities, which demonstrates the proposed method's generalizability to many application domains.

4. There is additional analysis showing that the proposed method works well with different amount of initially available data as well as varying budget, and also shows that balancing the budget across the selected modalities is better than giving more budget to some modalities than other in the selected subset.

**Weaknesses:**

While the main experiment is conducted on both affective datasets and ADNI/MIMIC, all the followup analysis about modality importance is on MOSI/MOSEI. The authors should at least provide the selected modalities for ADNI/MIMIC as well as their C(S) values.

The presentation quality of the paper needs improvement. There are many mistakes and inconsistencies within the figures and equations, including the following:

- In figure 1, instead of "left/right" in the caption, it should be "top/bottom" instead

- In Equation 1, {j} should be {$M_j$} instead on two locations (below the summation and at the union on the right), as j is just an index and $M_j$ is the modality element. Also, $\phi_j$ should be $\phi_{M_j}$ to be consistent with later equations.

- In Eq (1), the denominator should be (|M|-1)! instead of |M|!, right? Since S is a subset from |M|-1 total elements.

- In Eq(4), i is undefined. The {$M_i$} is probably a typo and should be {$M_j$} instead.

- Technically, $\hat{\phi}_S$ is not well-defined, as $\phi$ is only defined with single-element subscript in Eq (1) and (3). While I can understand how to extend $\phi$ calculation with a set S as subscript, the current mathematical expression is not very rigorous.

In this work, the authors assumed that the collection cost for each instance from each modality is equal. However, in the real world, the cost for collecting data from each modality will vary. It is unclear whether the proposed method can be extended to work with varying budget cost for different modalities.

**Questions:**

1. When given a certain budget and a large number of modalities, how do you determine the max number of modalities to sample (the hyperparameter k in Algorithm 1)? How do you balance the tradeoff between having more modalities versus having more data instances on fewer modalities?

2. It is quite strange that, in Table 1, quite a lot of upper bound values are lower than lower bound values. Do you have an explanation for this?

---

> ### Author Response · Authors · 2025-11-28
>
> We deeply appreciate the reviewer's strong endorsement and the thorough verification of our mathematical notation and figures. We are encouraged that you found the problem "interesting and important" and the solution "clever." We have carefully addressed the specific corrections and questions below.
>
> **1. Response to Analysis for ADNI/MIMIC**
>
> We have added the $C(S)$ analysis for the healthcare datasets in Table R1.
> - For ADNI, the indicator identifies the pair {Clinical, Biospecimen} (CB) as the most synergistic combination, achieving the highest accuracy. This suggests that in this limited-data regime, the inclusion of additional modalities like Genetics or Imaging to the robust CB core may introduce complexity or sparsity challenges that slightly dampen the synergy efficiency compared to the distinct complementarity found in the Clinical and Biospecimen pair.
> - For MIMIC, the combination {Text, Time-series} (LN) exhibits the highest positive cooperation, outperforming {Text, Codes} (LC). This indicates that the dynamic information in time-series vitals provides essential complementary context to the unstructured clinical text, whereas structured Codes (C) share more redundant information with the text.
>
> Table R1: The cooperation score and proposed indicator rank.
> | Datasets | S   | Acc. | C(S)  | Rank. |
> |----------|-----|------|-------|-------|
> | ADNI     | CB  | 61.8 | 0.374 | 1     |
> |          | IC  | 61.0 | 0.366 | 2     |
> |          | GC  | 58.0 | 0.333 | 3     |
> |          | GCB | 61.2 | 0.325 | 4     |
> |          | GB  | 53.7 | 0.280 | 5     |
> |          | IB  | 51.4 | 0.248 | 6     |
> |          | ICB | 52.2 | 0.208 | 7     |
> |          | IGC | 51.1 | 0.191 | 8     |
> |          | IGB | 49.8 | 0.170 | 9     |
> |          | IG  | 39.1 | 0.011 | 10    |
> | MIMIC    | LN  | 62.9 | 0.141 | 1     |
> |          | LC  | 61.2 | 0.118 | 2     |
> |          | NC  | 57.6 | 0.063 | 3     |
>
> **2. Response to Presentation and Math Errors**
>
> We are grateful for your careful review. We incorporate all these corrections in the final revision:
> - Fig 1 Caption: Corrected "left/right" to "top/bottom".
>
> - Eq 1 Notation: Corrected $\{j\}$ to $\{M_j\}$.
>
> - Eq 1 Denominator:We would like to clarify that although the summation is over subsets $S \subseteq \mathcal{M} \setminus \{j\}$ (which contains $|\mathcal{M}|-1$ elements), the Shapley value is defined as the average marginal contribution of modality $j$ over all possible permutations of the entire set of modalities $\mathcal{M}$.
>
> - Eq 4 Typo and Definition: fixed $i$ to $\{M_i\}$.
>
> - $\hat{\phi}_S$ Definition: We have revised the description for Eq 3-4 to separate individual contributions from subset interactions.
>
> **3. Response to Varying Costs in Data Collection**
>
> - We acknowledge that real-world acquisition costs vary across modalities. In this work, we formulated the budget $B$ as a sample count to focus on verifying the fundamental effectiveness of the Select-then-Generate paradigm in terms of information gain. We will consider exploring complex cost-aware modeling in specific real-world scenarios as future work.
>
> - Still, SK-ll is inherently extensible to varying costs: the current constraint ($\sum \beta_j \le B$) can be easily adapted to a weighted constraint ($\sum c_j \beta_j \le B_{total}$), where $c_j$ represents the specific unit cost of modality $j$.
>
> **4. Response to Questions**
>
> - Hyperparameter k: We do not treat $k$ (number of selected modalities) as a fixed hyperparameter to be tuned. Instead, Algorithm 1 is iterative. We stop increasing $k$ when the marginal gain (Improvement in $C(S)$) becomes negligible.
>
> - (Table 1) Lower Bound Setting Outperforms Upper Bound Setting Performance: This result is actually a key motivation for our paper. In multimodal learning, Modality Imbalance and Noise are significant issues. When we blindly add new data for weak or noisy modalities, it can confuse the model or increase the modality gap, leading to performance degradation (overfitting to noise). By selecting only the "Key" modalities, SK-II avoids this interference and consistently outperforms the lower bound, whereas naive data collection does not.

---

### Official Review · Reviewer_Bhsc · 2025-10-31

**Soundness:** 3
**Presentation:** 1
**Contribution:** 3
**Rating:** 4
**Confidence:** 5

**Summary:**

The paper proposes indicator to decide importance of key modality and generate the rest ones considering circumstances of limited data. Experiments on multiple multimodal downstream datasets are conducted to show effectiveness of the proposed model. Although the motivation makes sense for the paper, the presentation is poor for understanding.

**Strengths:**

1.	Modality importance indicators are theoretically solid with Shapley value computation.
2.	Experiments are thorough.

**Weaknesses:**

1.	The paper seems written in a rush and missing lots of theoretical and experimental details. For instance, the definition and calculation of Z in Equ. 3 and 4 as the core measuring function. Is it dataset-level or modality-level definition?s It’s confused to divide Z as normalization in two times. Besides, architecture of generator has not provided for diverse datasets or tasks. The presentation really need to be improved.
2.	Accuracy of UR-FUNNY and MUSTARD are not reported in Table 1 although the caption has mentioned it.
3.	As described in Section 3.4, the generation and optimization in each epoch contained multiple stage of training, which may increase heavy computation. The increased training efficiency compared with baselines should be reported in Table 10.
4.	Since the motivation is about data limitation, all division ratio from 10% to 100% of data budget should be reported in Table 2 to show consistent improvement of the proposed method.
5.	Ablation study should be remained as controlled variable method instead of the settings such as the selected modalities with diverse budget ratio in Table 4.
6.	Random selection of subset to compute C(S) in Table 11 should be reported with iterative experiments of nonoverlap selection across the whole dataset to ensure consistency.
7.	Visualization of generated and original modality embeddings should be conducted to show the generative effectiveness.
8.	More related paper or baselines should be discussed or compared with incomplete multimodal input. Miss Reference:
[1] 	Devamanyu Hazarika, Yingting Li, Bo Cheng, Shuai Zhao, Roger Zimmermann, and Soujanya Poria. 2022. Analyzing Modality Robustness in Multimodal Sentiment Analysis. In Proceedings of the 2022 Conference of the North American Chapter of the Association for Computational Linguistics: Human Language Technologies, pages 685–696, Seattle, United States. Association for Computational Linguistics.
[2] Ronghao Lin, Haifeng Hu; MissModal: Increasing Robustness to Missing Modality in Multimodal Sentiment Analysis. Transactions of the Association for Computational Linguistics 2023; 11 1686–1702. doi: https://doi.org/10.1162/tacl_a_00628
[3] Z. Yuan, Y. Liu, H. Xu and K. Gao, "Noise Imitation Based Adversarial Training for Robust Multimodal Sentiment Analysis," in IEEE Transactions on Multimedia, vol. 26, pp. 529-539, 2024, doi: 10.1109/TMM.2023.3267882.
[4] Adapt and explore: Multimodal mixup for representation learning, Information Fusion, 2024
[5] UniMF: A Unified Multimodal Framework for Multimodal Sentiment Analysis in Missing Modalities and Unaligned Multimodal Sequences, IEEE Transactions on Multimedia , 2024

**Questions:**

1.	Why reported Acc2 on MOSI while Acc7 on MOSEI in Figure 1. Besides, why are V+T+A reported as one edge of the pie while full modalities as a new pie area.

2.	In Equ 2, when there are no modality as input, does it mean to input all-zero vectors or Distribution sampled in normal Gaussian.

3.	Why not compute the S with single modality to better show the contribution of each modality and as supplementary evidences to support results in Table 1 in settings of S with two modalities.

4.	Why does some results in Table 2 show poorer than baselines?

5.	Why does some results with more data contribute less than other times in Table 2? Such as 20% to 30% compared with 30% to 40% ones on MIMIC.

---

> ### Author Response · Authors · 2025-11-28
>
> We sincerely thank the reviewer Bhsc for the thorough reading and for recognizing the theoretical solidity of our modality importance indicators. We appreciate the constructive feedback on presentation and experimental details, which we have addressed below.
>
> **1. Response to Theoretical and Experimental Details**
> We thank the reviewer for the detailed review. Some wording in Section 3.2 in early manuscript may have caused confusion. We have revised the manuscript to match our implementation.
> - Definition of $Z$: In our implementation, $Z$ corresponds to the absolute performance gain of the full multimodal model, i.e., $Z = f_u(\mathcal{M}) - f_u(\emptyset)$. Following [1], this scales $\phi_{M_j}$​​ relative to the total predictive capability of the model, reflecting the proportion of final performance attributable to modality $M_j$​. For the **Subset-level normalization $Z_S$**​ in Eq. (4). Our implementation uses the subset’s own absolute performance: $Z_S = f_u(S) - f_u(\emptyset)$. This normalizes the cooperation score $\mathcal{C}(S)$ by the capability of the subset itself, ensuring that the interaction reflects the fraction of performance due to cooperative effects rather than unimodal contributions. We have updated Section 3.2 to include these explicit definitions and removed the ambiguous reference.
> - Generator Architecture: For the MulT and CoMM backbones, we employ a class-conditional Normalizing Flow as the generator, adapting from [2]. In the case of Flex-MoE, we adhere to its original design by utilizing its missing modality bank mechanism to retrieve embeddings for missing data, rather than using an external generator. We have detailed the specific architectural hyperparameters in Appendix A3 of the revised paper.
> [1] Perceptual score: What data modalities does your model perceive?
> [2] Distribution-consistent Modal Recovering for Incomplete Multimodal Learning
>
> **2. Response to UR-FUNNY/MUSTARD Results in Table 1**
> Due to limited main paper space, in Table R1 we present the results from MOSI and MOSEI to illustrate the motivation behind our indicator design. We have provided the rest of the results in the table below. As observed, the rankings produced by our proposed indicator $\mathcal{C}(S)$ align with the actual model performance on these datasets as well.
>
> Table R1: The cooperation score and proposed indicator rank.
>
> | Datasets | S    | Acc.  | C(S)   | Rank. |
> |----------|------|-------|--------|-------|
> | UR-Funny | T, A | 58.95 | 0.155  | 1     |
> |          | A, V | 54.88 | 0.042  | 2     |
> |          | T, V | 45.50 | -0.186 | 3     |
> | MUStARD  | A, V | 62.45 | 0.085  | 1     |
> |          | T, A | 61.12 | 0.031  | 2     |
> |          | T, V | 57.45 | -0.045 | 3     |

---

> > ### Author Response · Authors · 2025-11-28
> >
> > **3. Response to Computation Efficiency**
> > - We appreciate the reviewer raising the point on computational overhead. We have updated Table 10 in the revised manuscript to include a detailed end-to-end model training time cost comparison. The results show that while SK-II introduces a pre-computation stage for the Indicator, this is a one-time, upfront investment. Besides, as demonstrated in our ablation study (Table 11), this indicator can be estimated using a small random subset (e.g., 100 samples), which reduces the pre-computation time on large datasets.
> > - Furthermore, our framework addresses the "Limited Data Budget" setting. In high-stakes domains like healthcare (ADNI/MIMIC) or affective computing, the bottleneck is the Data Acquisition Cost (e.g., expert labeling, intrusive sensors, patient recruitment), which is orders of magnitude higher than the cost of GPU compute time. SK-II trades a marginal increase in training compute (~15% for the generator) for a reduction in the need for expensive multimodal data collection. We believe this is a highly favorable trade-off for real-world deployment.

---

> > > ### Author Response · Authors · 2025-11-28
> > >
> > > **4. Response to Performance on Different Data Ratio Range**
> > >
> > > - We appreciate the reviewer's suggestion. As shown in Table R2, we extended our experiments to the 60%, 70%, and 80% availability ratios. From the table below, in the 60-70% range, SK-ll still provides consistent improvements over the Baseline. By the time availability reaches 80%, the performance of the model with either Baseline or SK-ll tends to saturate, especially for MUsTARD and ADNI.
> > > - This observation could be supported by our research motivation and setting, where the data is limited and new data collection budget constraints force a strategic trade-off between modalities. As the data availability approaches 100%, the problem of "selecting a subset" naturally becomes trivial, as the model converges to the full-modality upper bound.
> > >
> > > Table R2: Performance of SK-ll applied to more available data (Ava. D).
> > > | Ava. D | Gen. | Method   | MulT (MOSI) | MulT (MOSEI) | CoMM (UR-Funny) | CoMM (MUsTARD) | Flex-MoE (ADNI) | Flex-MoE (MIMIC) |
> > > |--------|------|----------|-------------|--------------|-----------------|----------------|-----------------|------------------|
> > > | 60%    | -    | -        | 78.43       | 80.92        | 56.12           | 59.87          | 59.83           | 66.47            |
> > > | 60%    | ✗    | Baseline | 79.76       | 81.54        | 58.42           | 62.13          | 61.21           | 68.08            |
> > > | 60%    | ✓    | SK-ll    | 82.14       | 82.83        | 60.17           | 62.93          | 64.38           | 71.46            |
> > > | 70%    | -    | -        | 81.18       | 82.09        | 58.76           | 61.18          | 63.22           | 71.24            |
> > > | 70%    | ✗    | Baseline | 82.07       | 82.63        | 60.18           | 62.72          | 63.88           | 73.53            |
> > > | 70%    | ✓    | SK-ll    | 83.82       | 83.71        | 61.63           | 63.44          | 65.42           | 75.12            |
> > > | 80%    | -    | -        | 83.13       | 83.22        | 60.48           | 62.17          | 64.97           | 74.83            |
> > >
> > >
> > > **5. Response to Ablation Study Design**
> > >
> > > We wish to clarify that our framework operates in two distinct steps before running multimodal models with generators: (1) Selection: Determining the optimal subset of modalities. (2) Allocation: Distributing the data budget within that selected subset. To address the reviewer's concern, we have refined the caption of Table 4 to focus strictly on Allocation Step. In this table, the "Modality Subset" is not a variable; it is a pre-condition fixed by the output of Selection Step. We treat the Budget Allocation Ratio as the single controlled variable. This setup allows us to observe how different budget distributions impact performance when the optimal modalities are already chosen.

---

> > > > ### Author Response · Authors · 2025-11-28
> > > >
> > > > **6. Response to Subset Sampling Results Consistency for Indicator Calculation**
> > > >
> > > > We thank the reviewer for the insightful suggestion regarding the robustness of our subset sampling strategies for indicator estimation. To ensure the consistency of $\mathcal{C}(S)$, we perform 5 iterative experiments using non-overlapping subsets (randomly sampling 100 distinct instances for each iteration from the training set). The aggregated results are presented in the Table R3, While the absolute values of $\mathcal{C}(S)$ naturally fluctuate, the estimated means remain close to the single-run snapshot. More importantly, the relative ranking of modality subsets ($\{T, V\} > \{T, A\} > \{A, V\}$) remained identical.
> > > >
> > > > Table R3: Indicator Scores C(S) on MOSI computed from all available data versus a small random subset of 100 samples (Avg. of 5 Non-overlap).
> > > >
> > > > | Subset | Acc. (All Data) | C(S) on All Data | C(S) on 100 (Avg. of 5 Non-overlap) |
> > > > |--------|-----------------|------------------|-------------------------------------|
> > > > | T, V   | 77.52           | 0.014            | 0.101 ± 0.018                       |
> > > > | T, A   | 75.02           | -0.015           | 0.062 ± 0.007                       |
> > > > | A, V   | 55.63           | -0.066           | -0.094 ± 0.005                      |
> > > >
> > > >
> > > > **7. Response to Visualization of Generated Embeddings**
> > > >
> > > > We appreciate the reviewer's suggestion to qualitatively evaluate the feature generation. In the revision, we have added Figure 3 to the Appendix, which presents a t-SNE visualization comparing the distributions of generated features against ground-truth real data for missing modalities.
> > > >
> > > > **8. Response to Missing References**
> > > >
> > > > We thank the reviewer for these valuable references. We have updated the "Related Work" section to cite and discuss references. We would also like to clarify that:
> > > >
> > > > -  While works like [1-2, 5] focus on enhancing model robustness against random, retrospective modality dropout (e.g., sensor failure during inference), ours SK-ll focuses on the prospective allocation of a limited data collection budget.
> > > >
> > > > - Methodological Connections: We have added a discussion linking our generation module to the concepts found in [3-4]. While [3] uses noise to simulate missing data for robustness, SK-ll employs a generative model guided by Shapley value selection to proactively "fill in" uncollected modalities, ensuring that the limited budget is spent only on the most informative data while still leveraging full-modality backbones.
> > > >
> > > > [1] Analyzing Modality Robustness in Multimodal Sentiment Analysis.
> > > >
> > > > [2] MissModal: Increasing Robustness to Missing Modality in Multimodal Sentiment Analysis.
> > > >
> > > > [3] Noise Imitation Based Adversarial Training for Robust Multimodal Sentiment Analysis
> > > >
> > > > [4] Adapt and explore: Multimodal mixup for representation learning, Information Fusion
> > > >
> > > > [5] UniMF: A Unified Multimodal Framework for Multimodal Sentiment Analysis in Missing Modalities and Unaligned Multimodal Sequences

---

> > > > > ### Author Response · Authors · 2025-11-28
> > > > >
> > > > > **9. Response to Questions**
> > > > >
> > > > > - Q1: for the larger MOSEI dataset, we report 7-class accuracy (Acc-7) because it offers finer granularity to distinguish the efficiency gains of Multimodal Learning (MML) compared to unimodal baselines, which is the primary focus of the analysis in Figure 1. Regarding the radar chart, while "V+T+A" and "Full Modalities" utilize the same input set, we plot "Full Modalities" (orange area) as a constant reference boundary. This design serves a visualization purpose to highlight the performance gap. We will drop the axis of full modality for clarity.
> > > > >
> > > > > - Q2: $G(\emptyset)$ represents a blind predictor (predicting the majority class or the dataset mean) without any input features. It serves as the baseline to calculate marginal contribution.
> > > > >
> > > > > - Q3: Single-modality values are calculated as an intermediate step (Eq. 3) We report the Cooperation Score $\mathcal{C}(S)$ as our objective is to quantify comprehensive model interaction, which is the decisive factor for the SK-ll allocation strategy.
> > > > >
> > > > > - Q4: In Table 2, SK-ll consistently outperforms the Baseline (uniform allocation of the same limited budget), validating the efficacy of our selective allocation strategy. However, the Upper Bound utilizes ground-truth data for all modalities, SK-ll relies on generated features for the unselected modalities. Generated features inherently contain noise and a modality gap compared to ground-truth distributions. The results show that SK-ll is competitive with Upper Bound proving its data efficiency rather than benchmarking capability.
> > > > >
> > > > > - Q5: This could be attributed to phase transitions in deep learning [6]. The significant performance gain from 20% to 30% on MIMIC likely represents a tipping point where the model acquires sufficient data density to model complex clinical cross-modal correlations. The smaller marginal gain from 30% to 40% indicates the model capacity begins to saturate relative to the complexity of the task.
> > > > >
> > > > > [6] Deep learning scaling is predictable, empirically.

---

### Author Response · Authors · 2025-12-04
**General Response**

We greatly appreciate the time and effort that all the AC/SAC/PC and reviewers have put into their reviews of our paper. We extend our thanks for the valuable comments and suggestions that helped us improve our paper. In addition to the pointwise responses below, we summarize the updates that have been made to our paper.

**Extra Experiments**

- *Expanded benchmarks:* To address the concern for broader validation by reviewers Bhsc and 6hZs, we extended indicator calculations to ADNI, MIMIC, UR-FUNNY, MUsTARD. The results show that the rankings produced by our proposed indicator align with actual model performance across diverse datasets.

- *Performance on varying data availability:* Addressing the suggestion by Bhsc, we report additional results at 60%-80% data availability ratios. SK-II provides consistent improvements over baselines until performance saturates at high data levels, validating its robustness across different budget constraints.

- *Indicator validation:* To address concerns regarding computational overhead, we conducted ablation studies estimating the indicator on many non-overlaping random subsets. The results show the indicator remains effective, significantly reducing pre-computation time.

- *Visualization of generated data:* We provide t-SNE visualizations in the Appendix to qualitatively validate the distribution of generated features against ground-truth real data for missing modalities, as suggested by the reviews.

**Paper Editing**

- *Related Work Updates:* We have enriched Related Work to incorporate suggested references and better position SK-II within the literature. We cited and discussed works on missing modality robustness to distinguish our prospective data collection focus from their retrospective robustness focus. Also, we added discussions on recent modality imbalance works 2 and clarified the distinction between inference-time pruning (their focus) and acquisition-time budget allocation (our focus).

- *Methodological Clarifications:* We revised the Methodology Section to rigorously define the normalization terms and improved the notation in Eq. 1 and Eq. 4. We also expanded Appendix A.3 to provide specific architectural details.

- *Problem Formulation:* We refined Section 3.1 to explicitly define "Budget" ($B$) as Data Acquisition Cost (sample count), hoping to resolve the concern raised by Reviewers RKhA and Bhsc01. We also clarified the "Select-then-Generate" rationale to distinguish our work from inference-time pruning methods.


Hope our pointwise responses below can clarify any confusion the reviewers have.

---

### Meta-Review · Area_Chair_qNei · 2025-12-29

**Summary:**

The paper addresses a highly relevant and practical problem, how to maximize multimodal learning performance under a constrained data acquisition budget, hence the authors proposed Select the Key, Then Generate the Rest. While I and several reviewers find the core motivation interesting and the proposed solution is innovative, the execution of the manuscript falls short of the conference standards. I am recommending rejection primarily due to significant issues with presentation and clarity that require a comprehensive revision. For example, reviewer Bhsc felt the paper appeared "written in a rush", citing missing theoretical details, confusing definitions for normalization terms, and a lack of generator architecture specifics, 6hZs identified numerous mathematical inconsistencies and errors in the figures and equations that impeded readability.

**Reviewer Concerns:**

The authors provided a detailed rebuttal that addressed several concerns, yet core issues remain unsolved:

Pros:
The authors successfully clarified the definition of the normalization term $Z$ and the distinction between data acquisition budget and computational budget, which was a major point of confusion for Reviewer RKhA. They also provided the missing experimental results for UR-FUNNY and MUSTARD. The additional ablation studies in Table 11 effectively addressed wGxh's concern about the computational cost of the Shapley indicator by showing it works with small sample sizes.

Outstanding:
The primary outstanding issue is the presentation quality. The fact that multiple reviewers (Bhsc, 6hZs) had to correct basic mathematical notation, figure captions, and request definitions for core terms suggests the paper was submitted prematurely. While the rebuttal provided the content to fix these issues, the manuscript itself needs a significant rewrite to be rigorous and self-contained. Furthermore, the confusion from Reviewer RKhA regarding the theoretical claims (optimization dynamics vs. information theoretic bounds) indicates that the paper's framing of its contribution needs to be much more precise to avoid misleading readers about beating the full set.

**Reviewer Scores:**

Based on the above issues, I think the reviewers would maintain their scores.

---

### Decision · Program_Chairs · 2026-01-26

Reject